# Optoplasmonic characterisation of reversible disulfide interactions at single thiol sites in the attomolar regime

Serge Vincent[1✉], Sivaraman Subramanian[1] & Frank Vollmer[1✉]

Probing individual chemical reactions is key to mapping reaction pathways. Trace analysis of sub-kDa reactants and products is obfuscated by labels, however, as reaction kinetics are inevitably perturbed. The thiol-disulfide exchange reaction is of specific interest as it has many applications in nanotechnology and in nature. Redox cycling of single thiols and disulfides has been unresolvable due to a number of technological limitations, such as an inability to discriminate the leaving group. Here, we demonstrate detection of single-molecule thiol-disulfide exchange using a label-free optoplasmonic sensor. We quantify repeated reactions between sub-kDa thiolated species in real time and at concentrations down to 100's of attomolar. A unique sensing modality is featured in our measurements, enabling the observation of single disulfide reaction kinetics and pathways on a plasmonic nanoparticle surface. Our technique paves the way towards characterising molecules in terms of their charge, oxidation state, and chirality via optoplasmonics.

[1] Living Systems Institute, School of Physics, University of Exeter, Exeter EX4 4QD, UK. ✉email: sv316@exeter.ac.uk; f.vollmer@exeter.ac.uk

Access to single-molecule reactions to determine the state of participating species and their reaction mechanisms remains a significant technological challenge. The application of fluorescent optical methods to investigate a single molecule's reaction pathway is often non-trivial. Sophisticated fluorescent labelling may not be available, while the temporal resolution is limited by photobleaching and transit times[1,2]. Monitoring reactions between molecules that weigh less than 1 kDa is further complicated by labels, as adducts can have severely altered reaction kinetics. Non-invasive optical techniques for studying the nanochemistry of single molecules have thus been elusive.

Thiol and disulfide exchange reactions are particularly relevant to the field of nanotechnology[3,4]. The reversibility of the disulfide bond has, for example, paved the way to realising molecular walkers and motors[5,6]. Bottom-up thiol self-assembled monolayers have shown potential as building blocks for sensors and nanostructuring[7]. The precise attachment/detachment of thiolated DNA origami has even extended to the movement of plasmonic nanoparticles (NPs) along an engineered track[8]. In nature, disulfide bonds are a fulcrum for cell biochemistry. Reactions that form these links usually occur post-translation, stabilising folding and providing structure for a number of proteins[9–11]. The cell regularly controls disulfide bonds between thiol groups, alternately guiding species through reduction and oxidation[12]. Redox potentials and oxidative stress in this context are reflected in the relative concentrations of thiols and disulfides[13].

Thiol/disulfide equilibria can be quantified in bulk, although often at the expense of high kinetic reactivity and the need for fluorescent or absorptive reagents to measure the exchange[14]. One such approach is an enzymatic recycling assay with 5-thio-2-nitrobenzoic acid absorbers capable of detecting thiols and disulfides down to 100's of picomolar concentrations[15]. This trades off quenching of thiol oxidation and exchange with the optimisation of reaction rates and the disruption of the thiol/disulfide equilibrium. As a disulfide bridge consists of two sulfur atoms that can interact with a thiolate (i.e. the conjugate base of a thiol), disulfide exchange is fundamentally intricate and the reaction branches for single molecules have yet to be fully characterised in the literature. Distinguishing leaving groups through a sensing element has so far been unachievable.

State-of-the-art sensors capable of transducing single-molecule interactions into optical[16–18], mechanical[19–21], electrical[22–24], or thermal[25] signals continue to emerge. Here we employ a label-free optoplasmonic system[26] that has the specific advantage of detecting individual disulfide interactions in solution. Due to the hybridisation between an optical whispering-gallery mode (WGM) resonator and localised surface plasmon (LSP) resonance of a NP, perturbations to an LSP are observed through readout of a WGM coupled to it[27–29]. One strategy we propose is to immobilise thiolates on a gold NP surface with a separate functional group. Following selective covalent binding, immobilised thiolates may participate in redox reactions while under non-destructive probing. Reactions between sub-kDa reactants are monitored in real time and at concentrations as low as 100's of attomolar, hence isolating for the disulfide chemistry of single molecules in vitro. Such reactions frequently result in abrupt changes in hybrid LSP-WGM resonance linewidth/lifetime—a surprising phenomenon that was considered unresolvable by WGM mode broadening or splitting[30–32]. We clarify in this study that disulfide linkages to bound thiolate receptors can exclusively affect the hybrid LSP-WGM resonance linewidth, beyond a

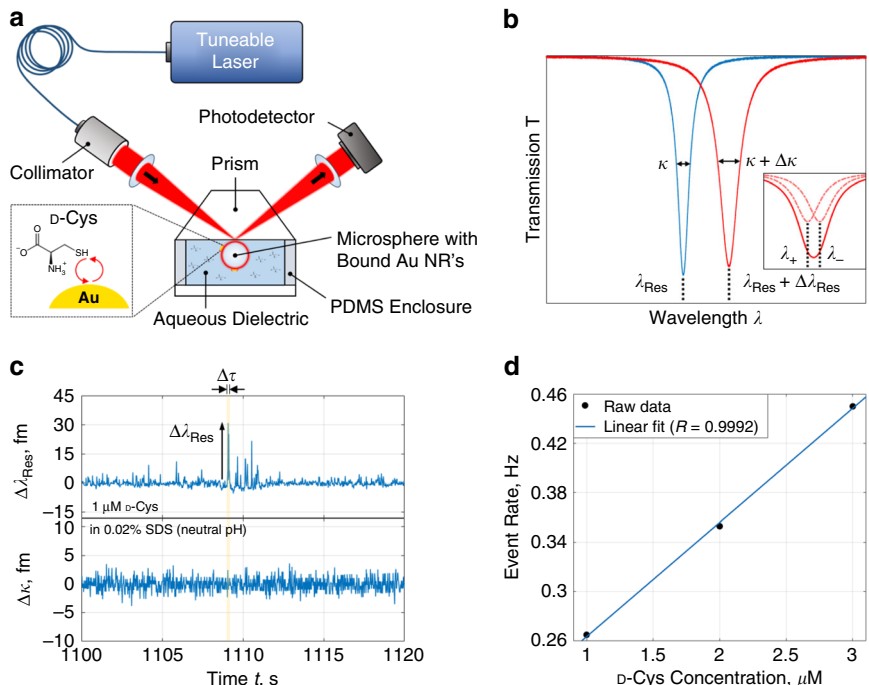

**Fig. 1 Optoplasmonic sensor setup and quantification of adsorbing D-cysteine. a** Scheme for LSP-WGM based sensing. A beam emitted from a tuneable laser source, with central wavelength of 642 nm, is focused onto a prism face to evanescently couple to a microspherical WGM cavity. The WGM excites the LSPs of Au NRs on the cavity surface and the hybrid system's transmission spectrum is acquired at the output arm of the setup. D-cysteine (D-Cys) analytes have carboxyl, thiol, and amine groups. **b** Sensing through tracking perturbations of the Lorentzian resonance extremum in the transmission spectrum. The resonant wavelength $\lambda_{\mathrm{Res}}$ and linewidth $\kappa$ that define the quality factor $Q = \lambda_{\mathrm{Res}}/\kappa$ are shown in the subfigure, as is unresolved mode splitting due to scattering. **c** Single-molecule time-domain signatures with signal value $\Delta\lambda_{\mathrm{Res}}$ and duration $\Delta\tau$ from the transit of D-Cys near Au NRs. The solvent used is 0.02% sodium dodecyl sulfate (SDS) in deionised water. **d** Linear dependence of event frequency on analyte concentration that suggests first-order rates. Events conform to a Poisson process (Supplementary Fig. 1).

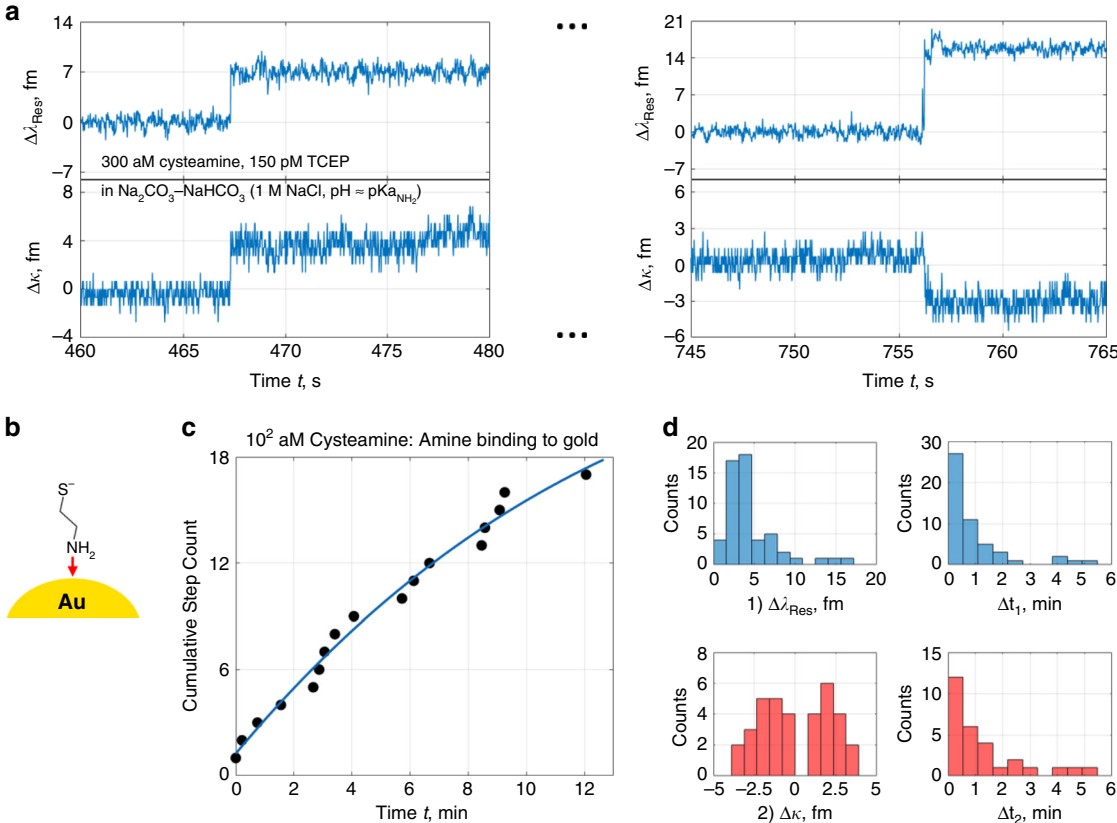

**Fig. 2 Single cysteamine binding to gold NRs via amine at subfemtomolar concentration. a** Discrete signals in the LSP-WGM resonance trace from covalent bonding of the $NH_2$ ligands to Au in a basic buffer. **b** Conceptual diagram of the cysteamine surface reaction. Cysteamine, with its thiol and amine groups, forms an amine-gold bond as indicated by the red arrow. **c** Exponential decay in cumulative binding step count as the system approaches saturation. In this regime, it is necessary to periodically inject more analyte in solution as scarce analytes are lost to external immobilisation (i.e. from undetected NRs that are not excited by the WGM). **d** Histograms depicting the resonance shift $\Delta\lambda_{Res}$ and linewidth shift $\Delta\kappa$ for binding events, as well as their related event time separations $\Delta t_1$ and $\Delta t_2$. The $\Delta\kappa$ distribution shows both positive and negative shifts, while $\Delta t_1$ and $\Delta t_2$ distributions are Poissonian.

description via an unresolved mode split. Each linewidth transition per exchange also assigns a status to the leaving group. Our data suggests a sensing modality for inferring kinetics and chains of single disulfide reactions in proximity to a plasmonic NP, paving the way towards assessing molecular charge, oxidation, and chirality states on an integrated platform.

## Results

**Experimental scheme**. A gold NP surface serves as an effective detection area for biomolecular characterisation on an optoplasmonic sensor. Light field localisation and nanoscale mode volumes at the NP hotspots enable sensitivity to surface modification, wherein covalent bonding to the NP restricts the total number of binding sites. Previously, thiol and amine based immobilisation has been explored on our optoplasmonic sensor[33]. Under particular pH conditions that dictate the molecular charge of the analyte, thiol and amine functional groups were reported to bind to different facets of gold NPs[34–37]. For thiols, the binding preference is in the (111) and (100) planes of a gold surface which are present in an ordered crystal lattice. For amines, binding preferentially takes place at uncoordinated gold adatoms. Measurements from[33] showed an approximate 2 orders of magnitude larger number of binding sites for thiols compared to amines on gold nanorods (NRs), demonstrating variable selectivity depending on surface regularity. If molecular charge is controlled and the NR surfaces are appropriately deformed, conditions can be reached where molecules containing both

amine and thiol groups can predominantly bind onto gold via amine to create recognised thiolates[38]. These nucleophiles may attack disulfide bonds in molecules that diffuse to them. Reducing agents introduced in solution, such as tris(2-carboxyethyl)phosphine (TCEP), can then reduce bound disulfides and complete a redox cycle. This pathway establishes cyclical reactions near the NP surface to be analysed statistically.

The LSP resonance of a plasmonic NP can be weakly coupled to a WGM resonance of a dielectric microcavity. Through this coupling, molecules that successfully perturb the gold NP surface can be detected as shifts in a LSP-WGM resonance. Light coupled in and out of the hybrid system allows for evaluation of gold NP perturbations, i.e. by laser frequency sweeping across the LSP-WGM resonance and spectrally resolving the resonant lineshape of the transmitted light. In our setup we excite WGMs in a silica microsphere, with diameters in the range of 70–90 μm, using a tuneable external cavity laser with 642-nm central wavelength. The laser beam is focused onto a prism surface to excite WGMs by frustrated total internal reflection. With a sweep frequency of 50 Hz, the transmission spectrum is acquired through photodetection at the output arm every 20 ms and a Lorentzian-shaped resonance is tracked (Fig. 1a, b). The evanescent field at the periphery of the microcavity is subsequently enhanced and localised in the vicinity of bound, LSP-resonant gold NRs. The cetyltrimethylammonium bromide-coated NRs have a 10-nm diameter and 24-nm length with longitudinal LSP resonance at $\lambda_0 = 650$ m. In the event of molecules interacting with the gold NR, the LSP-WGM lineshape position $\lambda_{Res}$ and/or full width at half maximum $\kappa$ will vary.

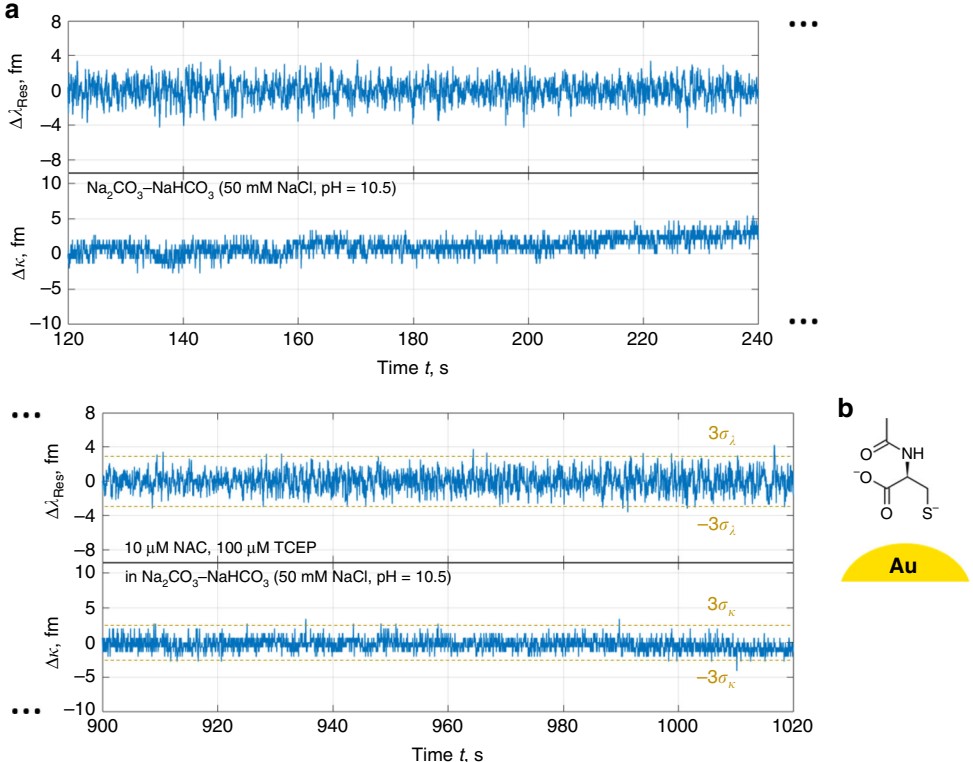

**Fig. 3 Background and negative control measurement with NAC at micromolar concentration. a** Resonance and linewidth shift traces exhibiting transient signal above $3\sigma$ with rates on the order of $0.1\,\text{s}^{-1}$ over several minutes; however, these persist in the presence and absence of NAC and TCEP in solution. No permanent binding patterns were found during peak tracking. **b** NAC molecule, with carboxyl, thiol, and (amine-attached) acetyl groups, near a detection site.

Discrete jumps in these parameters may be measured in the time domain and are indicative of molecular bond formation with gold. Groupings of signal fluctuations exceeding $3\sigma$ from transient arrival/departure can also arise (Fig. 1c), where $\sigma$ is the standard deviation of the noise derived from a representative 20-s trace segment. The resonance shifts of these signal packets are compiled for a series of analyte concentrations to confirm Poissonian statistics and first-order reaction rates (Fig. 1d). An extrapolation error exists in Fig. 1d given the chosen concentration range yet the event rate is most closely linear with concentration. Despite the negligible scattering and absorption cross-section of a single molecule, the ultrahigh-quality-factor WGM and its back-action on the perturbed LSP acts as a channel to sense loss changes intrinsic to or induced by a gold NP antenna. NP absorption spectroscopy by means of optoplasmonics[39] provides groundwork for such a modality, as the absorption cross-section change in a NP due to surface reactions may become detectable. We affirm that signal traces can exhibit (1) simultaneous shifts in resonant wavelength, linewidth, and resolved mode splitting[30,32] and (2) exclusive linewidth variation when single molecules diffuse within the LSP evanescent field decay length of the NP. Note here that the spectral resolution of our system is set by the laser frequency noise.

**Disulfide reaction mechanism and statistical analysis.** Loading of the gold NR surface with thiolate linkers requires a set of restrictions on the solvent environment at room temperature. To promote amine-gold bonds, we use a buffer at a pH that is above an aminothiol's logarithmic acid dissociation constants $\text{pKa}_{\text{SH}}$ and $\text{pKa}_{\text{NH}_2}$. Within this balance, anionic species with negatively charged $S^-$ and neutral $NH_2$ groups will dominate as per the Henderson–Hasselbalch equation[40]. A molecule must first reach

the gold surface by overcoming Debye screening from surface charges[41], e.g. from the gold NR's coating and pre-functionalisation of the glass microcavity. Such electrostatic repulsive forces can be reduced by electrolyte ions in substantial excess of the molecules under study. Analogous to raising the melting temperature of DNA from ambient conditions by increasing the salt concentration, the arrival rate of molecules to detection sites plateaus when the salt concentration is on the order of 1 M. Due to indiscriminate attachment of gold NRs onto the glass microcavity in steps preceding single-molecule measurements (Supplementary Fig. 2a), molecules in the medium should also be replenished to account for capture by NRs outside of the WGM's evanescent field (i.e. those that do not contribute to LSP-WGM hybridisation). Overall, these factors necessitate high electrolyte concentrations and recurring injection of analyte into a buffer of pH > $\text{pKa}_{\text{NH}_2}$ to attain a sufficient reaction rate in the subfemtomolar regime.

The aminothiol linkers of interest for our experiments are chemically simple amino acids or pharmaceuticals with minimal side chains. For chiral studies, D- and L-cysteine ($\text{pKa}_{\text{SH}} = 8.33$ and $\text{pKa}_{\text{NH}_2} = 10.78$[42]) are good candidates as they contain a carboxyl group that does not interfere with disulfide reactions. Nevertheless, for simplicity, we began with cysteamine ($\text{pKa}_{\text{SH}} = 8.19$ and $\text{pKa}_{\text{NH}_2} = 10.75$[43]) as it is a stable aminothiol that excludes any side chains. The cysteamine's amine group favourably binds to our optoplasmonic sensor in a sodium carbonate–bicarbonate buffer at a pH slightly above 10.75, with 1 M sodium chloride (Fig. 2a, b). Typical signal patterns in Fig. 2a for amine-gold binding are discontinuous steps in both $\lambda_{\text{Res}}$ and $\kappa$ on the order of 1 to 10 fm, with monotonic redshifts in $\lambda_{\text{Res}}$. Signal magnitude and direction depends on variables such as the position and orientation of the gold NR detector on the

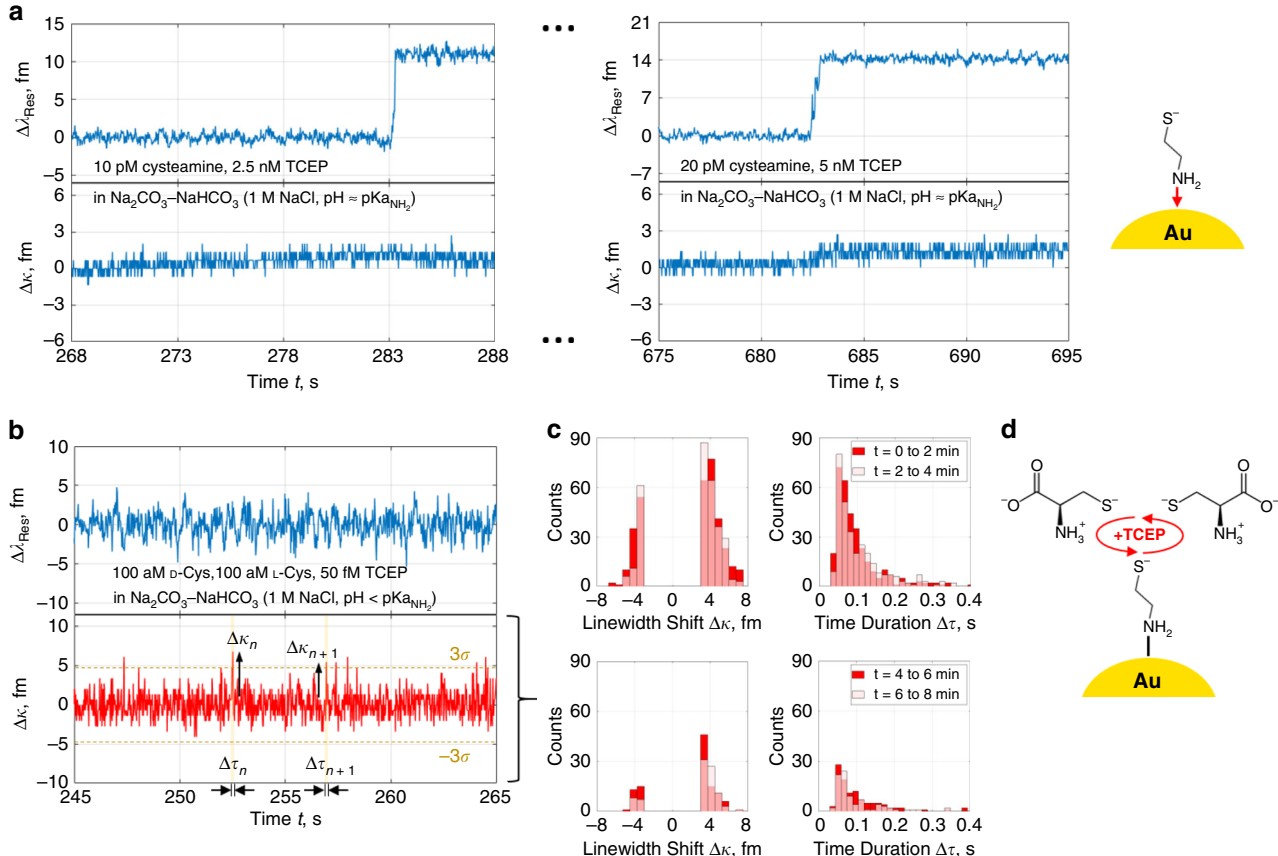

**Fig. 4 Cysteamine pre-functionalisation and disulfide events from converging DL-cysteine. a** Binding of cysteamine to Au NRs via amine in basic buffer. **b** Linewidth fluctuations induced by racemic DL-cysteine interacting with immobilised cysteamine thiolates at $pKa_{SH} < pH < pKa_{NH_2}$. TCEP reducing agent is employed here to counteract cysteine oxidation/dimerisation. **c** Linewidth shift $\Delta\kappa$ and event time duration $\Delta\tau$ histograms extracted from the resonance trace of (**b**). The mean event rate of the Poisson distributions passed through an inflection point, decreasing from 0.01 $aM^{-1}s^{-1}$ to 0.003 $aM^{-1}s^{-1}$ within an 8-min interval as the diffusing cysteines were captured. **d** DL-cysteine and bound cysteamine transiently interacting via their thiol groups.

microcavity[26], the detection site on the NR itself, and the analyte's molecular mass/polarisability[44]. As time evolves and analyte is steadily supplied, the binding sites become occupied and event rate decreases (Fig. 2c). These independent shifts in $\lambda_{Res}$ and $\kappa$ are collected in Fig. 2d to showcase non-monotonic linewidth narrowing and broadening once single molecules bind. This is an unconventional result as there are equally likely signs for $\Delta\kappa$ without apparent proportionality to $\Delta\lambda_{Res}$. A singlet of an unresolved mode split that would generate the linewidth shift is thus unsubstantiated.

An added convenience of choosing cysteamine is its comparable diffusion kinetics with respect to N-acetylcysteine (NAC)—a synthetic precursor of cysteine with acetyl protecting group in place of primary amine. We used NAC as a negative control and the response revealed a negligible rate of thiol-gold bond formation at high pH and high concentration (Fig. 3). A lack of step discontinuities within the trace supports amine-gold bonding in the basic buffer and therefore thiol-functionalisation of the gold NRs with cysteamine.

**pH-dependent disulfide nanochemistry.** The charge of single molecules diffusing to the optoplasmonic sensor can lead to a diverse set of reactions and LSP-WGM resonance perturbations. Dimerisation, for instance, is maximised when thiol groups are made nucleophilic through deprotonation at a pH above the $pKa_{SH}$. To circumvent electrostatic repulsion between primary amines, high aminothiol dimerisation and disulfide exchange

rates demand a pH greater than the $pKa_{NH_2}$. We therefore investigated these effects by way of pH variation near the $pKa_{NH_2}$. After pre-loading the gold NRs on the glass WGM microcavity with cysteamine in Fig. 4a, we flushed the chamber volume and replaced the surrounding dielectric with sodium carbonate-bicarbonate buffer, 1 M sodium chloride, at pH 10.19 < $pKa_{NH_2}$. Figure 4b highlights signal activity upon addition of a racemic, subfemtomolar mixture of reduced D- and L-cysteine. Transient peaks in the linewidth appear in packets that dissipate (see Fig. 4c) as external capture removes available DL-cysteine. We attribute these peaks to thiolates that fail to form a disulfide bond (Fig. 4d). The Poisson-distributed events for $t \leq 2$ min. have a mean rate = 0.01 $aM^{-1}s^{-1}$ that surpasses diffusion (i.e. $D_{DL-Cys} \sim 10^{-10}$ $m^2 s^{-1}$, $k_{on} \sim 1$ $nM^{-1}s^{-1}$ [45]), implying molecular trapping near the gold NR hotspots. Charged molecules are, by analogy to atomic ions[41], bounded by an electrostatic potential well whose depth is increased in proportion to ionic strength.

For proof of principle, we increased the environmental pH to 11.09 and raised the analyte concentration. In this regime we expected sustained reversible disulfide reactions with defined signal states in the resonance trace. The neutral amines of the highly anionic cysteamine and L-cysteine indeed result in binding/unbinding state transitions as in Fig. 5a, with clear linewidth broadening and narrowing steps of roughly equal mean height. The stability of the disulfide reactions is attributed to an order of magnitude rise in hydroxide ion concentration past the $pKa_{NH_2}$ and the event rate is maintained by electrostatic trapping.

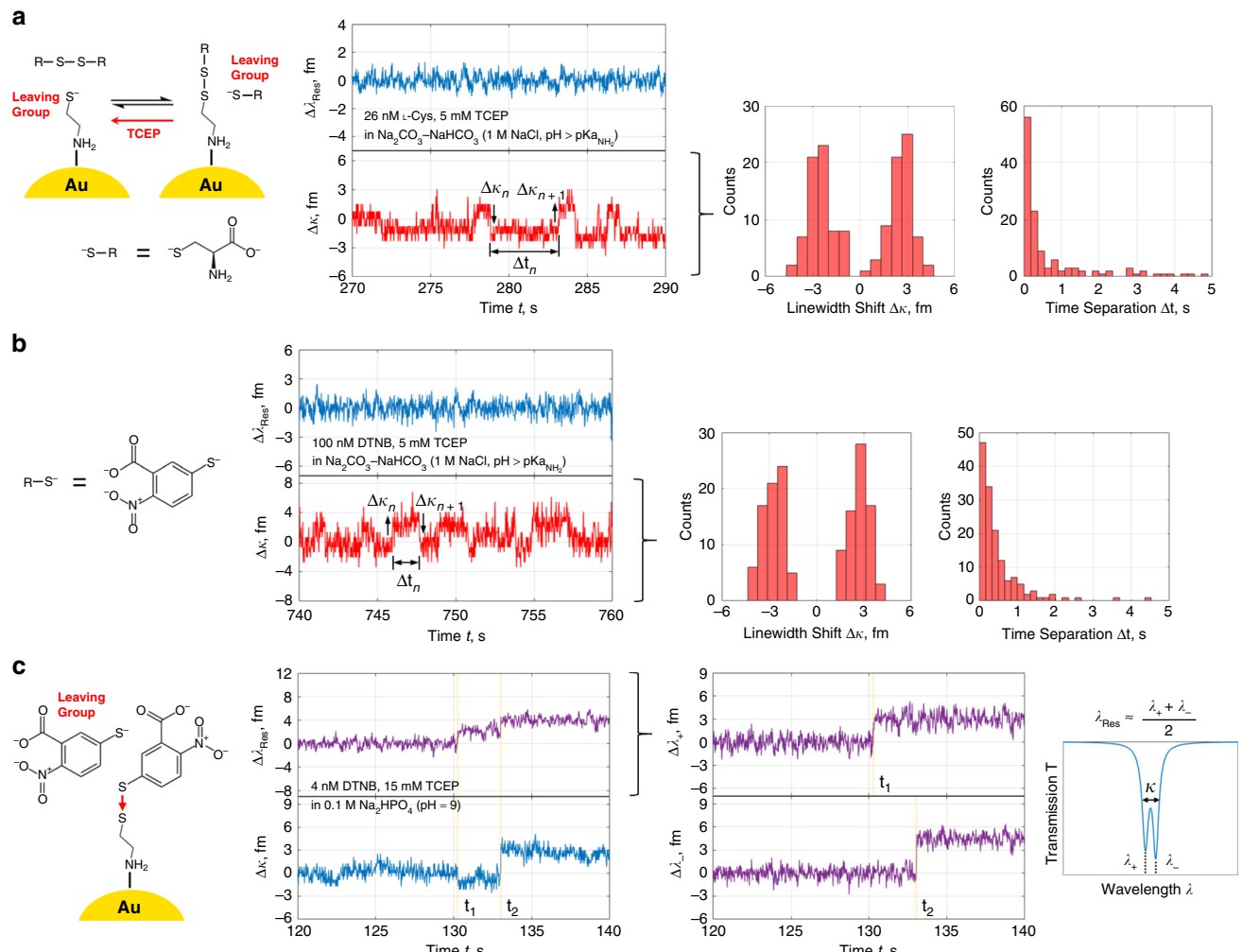

**Fig. 5 Cyclical binding/unbinding and exchange interactions with single mixed disulfides. a** Real-time linewidth step oscillations in the LSP-WGM resonance trace from redox reactions involving individual cysteamine-L-cysteine disulfides at pH > pKa$_{NH_2}$. These bridges are formed between cysteamine linkers and L-cysteine thiolates/disulfides (with neutral amines), then promptly cleaved by excess TCEP. **b** Linewidth patterns similar to **a** from individual cysteamine-TNB disulfides. Thiol-disulfide exchange may be triggered by DTNB dimers alone; however, cycling is ensured through reduction with TCEP. TNB has a benzene ring with carboxyl, thiol, and nitro groups. **c** Apparent resonant wavelength and linewidth signal steps, from thiol-disulfide exchange with DTNB and bound cysteamine, in a resolvable LSP-WGM doublet/split mode.

Since TCEP continually cleaves bound dimers during redox cycling, the monomer or dimer state of the leaving group can also be identified (cf. Supplementary Fig. 4). This trial was repeated in Fig. 5b for a larger molecule, 5,5′-dithiobis-(2-nitrobenzoic) acid (DTNB/Ellman's reagent), which readily underwent disulfide exchange with bound cysteamine linkers. In all cases, reducing agent concentration was adjusted until switching signals in the linewidth were observed. Resolvable dwell times and hence steady diffusion of reducing agent to the detection site were found at high molar excess > 1000.

Some insight into oscillation patterns is provided by the mode split traces of Fig. 5c, the lineshapes for which are discernible when the coupling/scattering rate is larger than the cavity decay rate. The WGM eigenmode degeneracy is lifted here and the resonant wavelength traces for the high-energy and low-energy modes, respectively denoted as $\lambda_+$ and $\lambda_-$, disclose two separate binding events in time. Such divergence comes from perturbations of two distinct gold NRs lying at different spatial locations along the standing wave formed by counterpropagating WGMs. One NP is excited near a node of a constituent mode and the second lies near in an antinode, and then the situation inverts for the other constituent mode. Information in the split mode

resonance wavelengths is encoded in the linewidth trace during single peak tracking; a shortcoming that, if corrected by available splitting, offers more robust molecular analysis by correlation to split mode properties and further detection site discrimination. Anomalous linewidth signatures of Fig. 5a, b that exclude resonance wavelength shifting are, however, only superficially explained via mode splitting. In order for the resonant wavelength to stay constant and relative mode splitting to be a contributing factor, either $\Delta\lambda_+ = -\Delta\lambda_-$ or the transmission dip depth must oscillate—two features that we have not detected in our recorded split mode traces. For the former to hold true, any heterodyne beat note tied to frequency splitting would have to stably oscillate between two beat frequencies. It is instead conceivable that the combination of LSP-WGM resonance energy invariance and lifetime variance implicates a relationship between the LSP resonance and molecular vibrational modes[46]. A transition between bound vibrational states that are close in energy and reside in two continuums is possible. With shifts in the electronic resonance-dependent Raman cross-section upon chemical reaction and/or charge transfer, the Raman tensor and hence the optomechanical coupling rate may be decipherable. In this way the charge state of bound cysteamine linkers and their

disulfide linkages can influence the optoplasmonic sensor response to grant molecular charge sensitivity[47].

## Discussion

Experimental results were presented for single aminothiols binding to gold nanoantennae of an optoplasmonic sensor system at subfemtomolar concentrations. We leveraged these aminothiol linkers (i.e. cysteamine) by way of reaction of their amine groups with gold, followed by repeatable disulfide interactions between the linkers and diffusing thiolates/disulfides incorporating TCEP reducing agent as a counterbalancing reagent. The thiol-functionalisation of gold was reinforced by negative controls performed with thiolated molecules in an equivalent sensor configuration. Statistical analysis of signal patterns at 100's of attomolar concentration revealed finite single-molecule detection due to removal from external adsorption, ligands, or other forms of capture. This recent advance is in part guided by selection of low-complexity analytes and saturation of environmental conditions to suppress Debye screening. Signatures in the linewidth traces were championed throughout our measurements as they were shown to contain leaving group information imprinted onto LSP-WGM resonance perturbations.

Despite the existence of identifiable disulfide interactions from DTNB, D-cysteine, and L-cysteine, a comprehensive theory to describe the underlying optoplasmonic detection mechanism has yet to emerge. Nonetheless, the dwell times and statistical inferences of cyclical single-molecule interactions in this work remain critical in circumventing site heterogeneity and characterising surface-bound thiolates and disulfides. Reactions near the nanoantennea hotspots have demonstrably lower degrees of freedom via spatial constraints and redox cycling. We foresee future refinements to the temporal resolution by locking the laser frequency to the WGM resonance. Our disulfide quantification paradigm ultimately opens avenues for charge transfer observation, including direct implementation of all sensing channels towards pinpointing single molecules and unravelling their nanochemistry.

## Methods

**Sample and microsphere preparation**. Chemicals were purchased from Sigma-Aldrich and Thermo Scientific. The principal solvent in which analytes were dissolved was ultrapure water delivered from a Merck Q-POD dispenser. Solutions without NRs were passed through a 0.2 μm Minisart syringe filter and dilutions were performed with Gilson P2L, P20L, and P1000L pipettes. Each microspherical cavity was reflowed using a Corning SMF-28 Ultra, single-mode telecommunications fibre by $CO_2$ laser light absorption. Surface tension during heating yielded a circularly symmetric cavity structure with a smooth dielectric interface. Mechanical stabilisation of the suspended microcavity was provided by prior insertion into a Thorlabs CF126-10 ceramic ferrule, which was then secured to an aluminium holder fixed to a three-axis translation stage. The diffusion-limited sample volume of 300–500 μL was enclosed by a glass window, N-SF 11 prism face, and sandwiched polydimethylsiloxane (PDMS) basin.

**Surface chemistry protocol**. Once the cavity was submerged in aqueous solution and a coupling geometry was found via alignment, cetyltrimethylammonium bromide-coated gold NRs (diameter = 10 nm, length = 24 nm, and LSPR wavelength = 650 nm) from Nanopartz were deposited onto the microcavity surface. A desirable linewidth change $\Delta\kappa$ accumulated during deposition was roughly 40–60 fm. Microsphere surface functionalisation and passivation are further detailed in Supplementary Methods 2. All aminothiol linkers were bound to the gold NRs in sodium carbonate-bicarbonate buffer at a pH above 10.75 with 1 M of sodium chloride ions. Additionally, washing steps were interspersed throughout each experiment to expel extraneous adsorbents.

**Resonance tracking**. In experiment, the whispering-gallery mode resonance extremum of our sensor is monitored using a bespoke centroid method[41,48]

$$\text{First Moment} = \frac{\sum_{i=1}^{n} i[T_{\text{Threshold}} - T(i)]}{\sum_{i=1}^{T} [T_{\text{Threshold}} - T(i)]}, \tag{1}$$

where $T_{\text{Threshold}}$ is the fixed transmission threshold and $n$ is the number of points defined to be in the resonant mode. The external cavity laser is swept linearly

across an ~8.5 pm wavelength range as driven by a triangular scan waveform, wherein hysteresis is averted by selective recording of the upscan. The transmission spectra are acquired with a sampling rate of 2.5 MHz and bit depth of 14. Given that laser diode emission intensity differs over the wavelength scan, 200 spectra are first averaged prior to coupling. Flattening of the spectrum is then executed and a fixed transmission threshold for peak detection is set. A resonance dip is only recognised if it falls below the transmission threshold and its width exceeds a successive point minimum. If these conditions are satisfied, the time trace of the computed lineshape position and width can be visualised in real time and stored for post-analysis. Many noise sources in the frequency domain are also put into account during our measurements, e.g. temperature drift (i.e. thermorefractivity and thermoelasticity), mechanical vibrations, laser mode hopping, and nanorod displacement.

## Data availability
The data that support the findings of this study are available from the corresponding author upon reasonable request.

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

## Acknowledgements

The authors acknowledge funding from the University of Exeter, the Engineering and Physical Sciences Research Council (Ref. EP/R031428/1), and from the European Research Council under an H2020-FET open grant (ULTRACHIRAL, ID: 737071). Spectral data was acquired and step signals were evaluated using LabVIEW software developed by M.D. Baaske.

## Author contributions

S.V. designed and performed the experiments, completed the data analysis, and composed the manuscript. S.S. wrote the MATLAB application for transient signal analysis, while F.V. supervised the project and revised the manuscript. All authors discussed and interpreted the results.

## Competing interests

The authors declare no competing interests.

## Additional information

**Peer review information** *Nature Communications* thanks the anonymous reviewers for their contribution to the peer review of this manuscript. Peer review reports are available.

