## [Peer Review File · Nature Communications]

REVIEWERS' COMMENTS:

Reviewer #1 (Remarks to the Author):

The authors have addressed all of my concerns except for one. Once the minor issue below has been resolved, I would recommend publication.

In their response to comment 1-5, the authors acknowledge that extrapolation error exists for a variety of reasons, and that extrapolation error would be lower with a wider concentration range. Yet, the authors only have 3 points and a relatively narrow concentration range. Consequently, I request that the authors acknowledge, directly in the manuscript, when they claim first order kinetics, that extrapolation errors exist and that they cannot definitively rule out a higher order mechanism.

Reviewer #3 (Remarks to the Author):

I thank the authors for answering my questions and clarifying the manuscript. I find the manuscript suitable for publication in Nature Comm.

The authors would like to thank the Reviewers for their input. In response to their comments, we have revised the main Manuscript. The revisions are outlined below with reference given to the relevant sections (indicated in **bold**).

Response to Reviewer 1:

Comments:

1-1) The authors have addressed all of my concerns except for one. Once the minor issue below has been resolved, I would recommend publication.

In their response to comment 1-5, the authors acknowledge that extrapolation error exists for a variety of reasons, and that extrapolation error would be lower with a wider concentration range. Yet, the authors only have 3 points and a relatively narrow concentration range. Consequently, I request that the authors acknowledge, directly in the manuscript, when they claim first order kinetics, that extrapolation errors exist and that they cannot definitively rule out a higher order mechanism

Response 1-1

We have made changes on **Lines 112-113** and **309-310** within the main Manuscript. We now state, in the presence of extrapolation errors, that the event rate vs. concentration curve is most closely linear as to suggest first order kinetics (i.e. in **Figure 1**'s legend).

Response to Reviewer 3:

Comments:

3-1) I thank the authors for answering my questions and clarifying the manuscript. I find the manuscript suitable for publication in Nature Comm.

Response 3-1

We thank Reviewer 3 for their participation/contribution during the review of our manuscript.